# Vicarious Social Defeat Increases Conditioned Rewarding Effects of Cocaine and Ethanol Intake in Female Mice

**DOI:** 10.3390/biomedicines11020502

**Published:** 2023-02-09

**Authors:** Francisco Ródenas-González, María Carmen Arenas, María Carmen Blanco-Gandía, Carmen Manzanedo, Marta Rodríguez-Arias

**Affiliations:** 1Unidad de Investigación Psicobiología de las Drogodependencias, Departamento de Psicobiología, Facultad de Psicología, Universitat de Valencia, 46010 Valencia, Spain; 2Departamento de Psicología y Sociología, Facultad de Ciencias Sociales y Humanas, Universidad de Zaragoza, 44003 Teruel, Spain

**Keywords:** social stress, cocaine, ethanol, conditioned place preference, oral self-administration, female mice

## Abstract

Stress is a critical factor in the development of mood and drug use disorders. The social defeat model is not appropriate for female rodents due to their low level of aggression. Therefore, a robust female model of social stress needs to be developed and validated. The aim of the present study was to unravel the long-lasting effects of vicarious social defeat (VSD) on the conditioned rewarding effects of cocaine and ethanol intake in female mice. Although VSD seems to be a good model for inducing behavioral and physiologic endophenotypes induced by stress, there are no studies to date that characterize the effect of VSD on cocaine or alcohol use. The results confirm that VSD females showed an increase in corticosterone levels after a vicarious experience while also displaying an increase in anxiety- and anhedonic-like behaviors. Three weeks after the last VSD, vicariously defeated female mice showed an increased developed preference for a non-effective dose of cocaine in the conditioned place preference (CPP) paradigm and showed an increase in ethanol intake. Our results suggest that female mice vicariously experience a state of distress through the social observation of others suffering from adverse events, confirming the use of VSD as a valid model to study the response to social stress in females. The fact that VSD in females induced a comparable behavioral phenotype to that observed in physically defeated males could indicate a relationship with the higher rate of psychopathologies observed in women. Notwithstanding, more studies are needed to dissect the neurobiological and behavioral peculiarities of the female response to social stress.

## 1. Introduction

Studies published to date have demonstrated that stress is a critical factor in the development of mood disorders such as depression or anxiety [1]. For instance, the neurotransmitter serotonin has been closely associated with the appearance of psychiatric disorders such as depression and anxiety [2], but it is also deeply affected by the experience of stress [3]. Stress-induced 5-HT activity occurs in areas such as the ventral striatum [4], the hippocampus [5], or the cerebellum [6]. In addition, social stress increased the kynurenine pathway of tryptophan metabolism, with numerous studies linking the kynurenine pathway with neuroinflammation and depression [7,8]. Psychosocial stress factors have long been associated with drug abuse [9,10], especially in women. Negative affect and stress reactivity are key components in the development of an alcohol use disorder in women [11,12]. Moreover, women were more likely than men to have stress preceding cocaine use [13].

Although numerous preclinical models have been developed to study these effects of stress, the social defeat (SD) model is the most widely used. Since social conflict can be a precipitating factor for many mental illnesses, SD induces behavioral responses that model symptoms of mental illnesses, many of which persist long-term after the defeat experience [14,15]. This model is based on a species-typical aggression between males, consisting of an encounter between conspecifics of the same species, modeling the relations of subordination and power of human beings [16,17]. In response to repeated SD, two different populations are observed: a group of susceptible rodents that show reduced social investigation or sucrose preference and a group of resilient mice that behave as control non-stressed mice [18,19].

Among other reasons, the low level of aggression observed in female rodents entails that most of these studies have been carried out only in male animals, although the frequency of presentation of stress-related diseases, such as depression or anxiety, is twice as frequent in women [20]. To better characterize and treat mood disorders in women, a robust female model of social stress needs to be developed and validated. Considering that the estrous cycle phase does not modify social exploration, which is the main assay used to evaluate depression-like behavior, concerns about behavioral variability in female mice due to hormonal changes should disappear [21].

Several studies have been carried out using a strain of mice in which females show aggression in defense of their territory, despite SD only inducing social avoidance in females [22,23]. Another series of studies have been developed based on the induction of male aggression toward females, for instance, through the chemogenetic activation of the ventromedial hypothalamus [24], with only susceptible females displaying social avoidance or anxiety-like behavior. However, Yin and co-workers [25] induced SD in females using male *DREADD* aggressors, which showed similar behavioral, physiological, and immune responses to those reported in males.

All of the previously described procedures rely on exposure to physical stressors. These studies highlight the strong link between stress and depression-related behaviors but do not represent the nature of stress in humans, with a prevalence of social and emotional stressors. Emotional or psychological stress alone could by itself induce mood-related disorders [26,27]. The SD model can be modified to include a mouse that has a vicariously-observed stress condition [28]. When this vicarious SD (VSD) is witnessed by another male, these mice exhibit depressive-like behaviors such as that observed in physically stressed mice [29,30]. These results have also recently been confirmed for female mice [31]. After experiencing 10 VSD, female mice showed depressive-like behaviors with elevated corticosterone and decreased body weight. However, no changes were observed in anxiety-like behaviors, conversely to the results obtained in males [28]. For these authors, VSD is capable of differentiating the neurobiological factors underlying specific mood-related syndromes (anhedonia vs. anxiety) as a function of sex.

In addition to increased psychopathologies, numerous experiments in rodents prove that experiencing SD is associated with increased cocaine use. Stress is one of the main risk factors involved in drug use, playing a role not only in relapse but also in the initiation, escalation, and maintenance of drug use [32]. The close relationship between the brain systems involved in addiction and stress allows environmental stressors to induce long-term changes in the function of the brain reward system. SD resulted in increased intravenous self-administration (SA) of cocaine and increased cocaine-seeking behavior, even one month after their last SD [33,34,35]. Similarly, the craving and reinstatement of cocaine use are strongly triggered by exposure to stressful events such as SD [32]. The results obtained using the conditioned place preference (CPP) paradigm were, in general, similar to those obtained after SA [36,37].

Preclinical studies also show that exposure to SD increases alcohol intake compared to non-stressed animals [38]. Using the oral alcohol SA paradigm, an increase in consumption and motivation to obtain the substance has been observed in defeated animals, an effect that is maintained several months after the last SD [38,39].

To sum up, although numerous studies have undoubtedly confirmed the increased rewarding effects of cocaine and ethanol after exposure to SD on male mice, no studies have previously evaluated these effects on socially stressed female mice. In this study, we have employed a VSD model in which females were not physically attacked but witnessed a standard SD between two males and remained confined with the resident aggressive male for a period of 24 h.

### Aims

The aim of the present study was to unravel the effects of VSD on the cocaine-conditioned rewarding effects and ethanol intake in female mice, confirming that it induces an increase in corticosterone response. Changes in the proinflammatory interleukin-6 (IL-6) were also evaluated in the striatum after the last VSD episode and the cocaine-induced CPP. We have repeatedly shown that SD in male rodents induces a long-lasting increase in the conditioned rewarding effects of cocaine evaluated with the CPP [35,37,40]. Similarly, SD increased ethanol intake in mice 3 weeks after the last defeat measured with oral ethanol SA [38,41]. However, although VSD seems to be a good model for inducing behavioral (social avoidance, anhedonia, and despair) and physiologic (increased corticosterone and weight change) endophenotypes induced by stress, no studies have been performed to date to characterize the effect of VSD on cocaine or alcohol effects. In a recent study, Newman and co-workers [42] exposed female mice to a 10-day schedule of chronic SD stress induced by an aggressive resident female, followed by cohabitation with the aggressor 24 h post-defeat. Although this study showed defeated female mice drinking more alcohol than controls for 4 weeks after the SD, it cannot be considered to be about VSD, since females were physically exposed to an aggressive mouse. We hypothesized that females exposed to VSD would show a comparable phenotype to that of physically defeated male mice, with increased corticosterone levels, higher levels of anxiety and depressive-like behaviors. We also hypothesized that the effect of VSD will last in time, as vicariously defeated females will develop a preference for a non-effective dose of cocaine and will drink more ethanol than non-exposed female controls 3 weeks after the last exposure to VSD.

## 2. Materials and Methods

### 2.1. Study Design

A total number of 131 OF1 mice (77 females and 40 males) on postnatal day (PND), 42, acquired from Charles Rivers (France) were used in this study. Experimental mice were housed in groups of four in plastic cages (28 × 28 × 14.5 cm) during the experimental procedure, except during the VSD. Adult males (N = 20) were used as aggressive opponents and were individually housed in plastic cages (21 × 32 × 20 cm) for at least a month prior to initiation of the experiments to heighten aggression [43]. Another 20 adult male mice were housed in groups of 4 (cage size: 28 × 28 × 14.5 cm) under standard conditions to be used as intruders.

All mice were kept in controlled laboratory conditions: constant temperature and humidity and a reversed light schedule (white light from 8:00 to 20:00). Food and water were available ad libitum to all the mice used in this study, except during behavioral tests. All procedures were conducted in compliance with the guidelines of the European Council Directive 2010/63/UE regulating animal research and were approved by the Local Ethics Committees of the University of Valencia for the use of animal subjects (Comité d’Ética d’Experimentació i Benestar Animal, number 2020/VSC/PEA/0082).

In this study, two different sets of mice were employed, both of which were exposed to the VSD procedure from PND 47 to 56. A more detailed description of the experimental design is illustrated in Figure 1. 

In the first set of mice (66 experimental females and 20 males for the VSD procedure), 30 min after the first and the fourth VSD sessions, blood sampling for corticosterone determination was performed in control (n = 8) and VSD (n = 8) females. A similar procedure was carried out 4 h after the first and fourth VSD sessions using other female mice (control, n = 8; VSD, n = 8). Additionally, we studied whether mice presented an inflammatory response; for this, biological samples were taken after the fourth VSD session (n = 14) and the CPP protocol (n = 16). A total of 24 h after the last VSD episode on PND 57, all the females performed the Social Interaction Test (SIT) to evaluate depressive-like behaviors, the Elevated Plus Maze (EPM) to test anxiety on PND 58, and the Splash Test (ST) and Tail Suspension Test (TST) to evaluate anhedonia-like behaviors on PND 59. Three weeks after the last VSD, the first set of female mice underwent the CPP procedure with 1.5 mg/kg of cocaine on PND 77-84.

In the second set of mice (20 experimental females and 20 males for the VSD procedure), three weeks after the last VSD episode on PND 56, all the females started the drinking in the dark (DID) test for four days, and, in the following week, the animals started the ethanol SA protocol for approximately 22 days (PND 77-113).

### 2.2. Drugs

For CPP conditioning, animals were injected intraperitoneally with a dose of 1.5 mg/kg of cocaine hydrochloride (Alcaliber laboratory, Madrid, Spain) dissolved in physiological saline (NaCl 0.9%) and adjusted to a volume of 0.01 mL/g of weight. This dose of cocaine was selected on the basis of previous CPP studies, showing that doses below 3 are sub-threshold [35,37,40]. Control groups were injected with physiological saline (NaCl 0.9%), which was also used to dissolve the drugs. For the DID and oral SA procedures, absolute ethanol (Merck, Madrid, Spain) was diluted in water using a 20% (*w*/*v*) ethanol solution.

### 2.3. Apparatus and Procedures

#### 2.3.1. Procedure of Vicarious Social Defeat (VSD)

The VSD paradigm was performed based on the previously described protocol [31]. Female OF1 mice vicariously experience the defeat of a male OF1 counterpart. In this protocol, VSD females were exposed to non-physical sensory stimuli (visual, olfactory, and chemosensory) associated with indirectly experiencing the defeat of the physically stressed male mouse. For each SD session (15 min/day), intruder male mice were placed into the same compartment as the aggressive resident, while VSD female mice were placed in the neighboring compartment, allowing only a vicarious experience (i.e., visual, olfactory, auditory) of the aggressive encounter. Females were exposed to four episodes of SD vicariously, and following each session, the female mouse stayed housed for 24 h with the resident, separated from the aggressive male mice with a perforated Plexiglas wall (31 × 18 × 0.6 cm) in between both areas. Females were physically protected from the male encounter but not from visual, olfactory, and auditory threats, which are part of the vicarious episode. After 24 h, the female was taken back to her home cage and their mates until the following encounter. The female control group underwent the same protocol but without the presence of a male SD encounter in the cage and without the presence of the resident mouse during the 24 h of housing.

#### 2.3.2. Determination of Plasma Corticosterone (ELISA) 

Blood sampling for corticosterone determination was performed by the tail-nick procedure, in which the animal is wrapped in a cloth, and a 2 mm incision is made at the end of the tail artery. The tail is then massaged until 50 μL of blood is collected in an ice-cold Microvette CB 300 capillary tube (Sarstedt, Nümbrecht, Germany). Blood samples were kept on ice, and plasma was separated from whole blood by centrifugation (5 min, 5000 g) and transferred to sterile 2 mL microcentrifuge tubes. Plasma samples were stored at −80 °C until the determination of corticosterone. On the day of the assay, samples were diluted (in a proportion of ~1:40) in the Steroid Displacement Reagent mix provided with the kit. Corticosterone levels in diluted plasma were then analyzed using a corticosterone EIA kit (Enzo Life Sciences, Catalog No. ADI-900-097, 96-Well kit), according to the manufacturer’s instructions, and an iMark microplate reader (Bio-Rad, Hercules, CA, USA) and Microplate Manager 6.2. software. The optical density was read at 405 nm, with 590 nm correction. The sensitivity of the test is 0.2.

#### 2.3.3. Procedure of the Social Interaction Test (SIT) or Social Withdrawal Ratio 

The social withdrawal ratio (SWR) used was based on the social approach-avoidance test previously described by Berton and co-workers [18]. The test took place 24 h after the last VSD during the dark cycle and in a different environment from the confrontation sessions. First, the animals were transferred to a quiet, dimly lit room 1 h before the test was initiated. After habituation, each animal was placed in the center of a square arena (white plexiglass open field, 30 cm on each side and 35 cm high), and its behavior was monitored by video (EthoVision XT 11, 50 fps; camera placed above the arena). Animals were allowed to explore the arena twice, for 600 s in each session, during two different experimental sessions. In the first session (object session), an empty perforated Plexiglas cage (10 × 6.5 × 35 cm) was placed in the middle of one wall of the arena. In the second session (social session), an unfamiliar male mouse was introduced into the cage as a social stimulus. Although it can be argued that the probe mouse used in the SIT resembles the aggressor and that this could foster social aversion, this is unlikely since previous experiments demonstrate similar amounts of social investigation, irrespective of the strain used (i.e., C57BL/6) [6]. Before each session, the arena was cleaned with a 5% alcohol solution to minimize odor cues. Between sessions, the experimental mouse was removed from the arena and returned to its home cage for 2 min.

Locomotion and arena occupancy during object and social sessions were determined using the animal’s horizontal position, determined by commercial video tracking software (EthoVision XT 11, Noldus, Wageningen, The Netherlands). Conventional measures of arena occupancy, such as time spent in the interaction zone and corners, were quantified. The former is commonly used as a social preference-avoidance score and is calculated by measuring the time spent in a 6.5 cm wide corridor surrounding the restraining cage. Corners were defined as two squares of similar areas on the opposite wall of the arena. The SWR is calculated by considering the time spent by an experimental mouse in the interaction zone when a social target is present divided by the time it spends in the interaction zone when the target is absent. A ratio equal to 1 means that equal time has been spent in the presence versus absence of a social target. Based on the regular behavior of control mice, animals with a ratio under 1 are classified as susceptible, while those with a ratio equal to or higher than 1 are classified as resilient [44].

#### 2.3.4. Elevated Plus Maze (EPM)

The elevated plus maze (EPM) test was carried out essentially following the procedure described by Ferrer-Pérez and co-workers [35,37]. The maze consisted of two open arms (30 × 5 × 0.25 cm) and two enclosed arms (30 × 5 × 15 cm), and the junction of the four arms formed a central platform (5 × 5 cm). The floor of the maze was made of black plexiglass, and the walls of the enclosed arms were made of clear plexiglass. The open arms had a small edge (0.25 cm) to provide the animals with additional grip. The entire apparatus was elevated 45 cm above floor level. In order to facilitate adaptation, mice were transported to the dimly illuminated laboratory 1 h prior to testing. At the beginning of each trial, subjects were placed on the central platform so that they were facing an open arm and were allowed to explore for 5 min. The maze was thoroughly cleaned with a damp cloth after each trial. The measurements recorded during the test period were the number of entries and time and percentage of time spent in each section of the apparatus (open arms, closed arms, and central platform). An arm was considered to have been visited when the animal placed all four paws on it. The time and percentage of time spent in the open arms and the number of open-arm entries are generally used to characterize the anxiolytic effects of drugs. In addition, the number of closed and total entries indicates motor activity.

#### 2.3.5. Splash Test (ST)

The splash test consisted of spraying a 10% sucrose solution on the dorsal coat of the mice, which were habituated to the room for 1 h before testing, following the protocol previously described by Hodes and co-workers [45]. The total time of grooming over a 5-min period was recorded and scored by blind-trained observers. The percentages of time spent by the animal’s leg- and back-grooming were analyzed. A decrease in grooming in the back of the animals can be considered a symptom of apathy, one of the behavioral endpoints for measuring the anxious-depressive phenotype in rodents [46].

#### 2.3.6. Tail Suspension Test (TST)

In the tail suspension test, each mouse was suspended individually (using adhesive tape attached 1 cm from the tip of the tail) 50 cm above a benchtop for 6 min. The time spent immobile by the animal during this interval was recorded and scored by an observer blind to the experimental conditions [47].

#### 2.3.7. Conditioned Place Preference (CPP)

For place conditioning, we employed eight identical plexiglass boxes with two compartments of equal size (30.7 × 31.5 × 34.5 cm high) separated by a central gray area (13.8 × 31.5 × 34.5 cm high). The compartments had different colored walls (black vs. white) and distinct floor textures (a fine grid in the black compartment and a wide grid in the white one). Four infrared light beams in each compartment of the box and six in the central area allowed the position of the animals and their crossings from one compartment to the other to be recorded. The equipment was controlled by three computers using MONPRE 2Z software (CIBERTEC, SA, Madrid, Spain).

Place conditioning, consisting of three phases, was carried out during the dark cycle following a procedure that is unbiased in terms of initial spontaneous preference [37]. During the first phase, or preconditioning (Pre-C), mice were allowed access to both compartments of the apparatus for 900 s per day on 3 consecutive days. On day 3, the time spent in each compartment was recorded. Animals showing a strong unconditioned aversion (<33% of session time; i.e., 250 s) or preference (>67% of the session time; i.e., 650 s) for any compartment were discarded from the rest of the study. The ANOVA showed no significant differences between the time spent in the drug-paired and vehicle-paired compartments during the Pre-C phase. In the second phase (conditioning), which lasted 4 days, animals were conditioned with 1.5 mg/kg cocaine or saline. During this phase, half of the animals in each group received the drug or vehicle in one compartment, while the other half received it in the other compartment. An injection of physiological saline was administered before confining the mice to the vehicle-paired compartment for 30 min. After an interval of 4 h, the animals received cocaine immediately prior to confinement in the drug-paired compartment for a further 30 min. The central area was made inaccessible by guillotine doors during conditioning. The dose of cocaine used during the conditioning phase was a subthreshold dose (1.5 mg/kg, proven to be ineffective in controls) in order to evaluate increased sensitivity to the conditioned rewarding effects of cocaine. In the third phase, or post-conditioning (Post-C), which took place on day 8, the guillotine doors separating the two compartments were removed, and the time spent in each compartment by the untreated mice during a 900-s observation period was recorded. The difference in seconds between the time spent in the drug-paired compartment during the post-C and pre-C tests is a measure of the degree of conditioning induced by the drug (conditioning score). If this difference is positive, then the drug has induced a preference for the drug-paired compartment, while the opposite indicates an aversion.

#### 2.3.8. Drinking in the Dark (DID)

Following the basic paradigm of Rhodes and co-workers [48], the test consists of two phases. The first is habituation, where the animals were removed from their cages to be housed individually for one week to habituate them to the cages and the suction tubes containing a ball bearing at the end to prevent leakage, which was used throughout the test. In the second phase of the protocol, the test began 3 h after lights out, and the water bottles were replaced with 10 mL graduated cylinders containing a 20% (*v/v*) ethanol solution. These remained in place for 2 h. After this 2 h period, the animals were returned to their grouped cages, with food and water bottles ad libitum again. This procedure was repeated on days 2 and 3, and on day 4, the procedure lasted for 4 h. In addition, immediately after each day, liquid consumption was recorded. Fresh ethanol solution was prepared each day. In our case, we maintained the protocol for one week for habituation to ethanol.

#### 2.3.9. Oral Ethanol Self-Administration (SA)

This procedure is based on that employed by Navarrete and co-workers [49]. Oral ethanol SA was carried out in 8 modular operant chambers (MED Associated Inc., Georgia, VT, USA). A software package (Cibertec, SA, Madrid, Spain) controlled the stimulus and fluid delivery and recorded operant responses. The chambers were placed inside noise isolation boxes equipped with a chamber light, two nose-poke holes, one receptacle to drop a liquid solution, one syringe pump, one stimulus light and one buzzer. Active nose-pokes delivered 20 μL of fluid combined with a 0.5 s stimulus light and a 0.5 s buzzer beep, which was followed by a 6 s time-out period. Inactive nose-pokes did not produce any consequence. 

To evaluate the consequences of VSD on the acquisition of oral ethanol SA, animals underwent an experiment carried out in three phases: training, fixed ratio 1 (FR1) and progressive ratio (PR) with a 20% ethanol concentration. In the training phase (12 days), mice were trained to respond to the active nose-poke to receive 20 μL of 20% (*v/v*) ethanol reinforcement. No food or water deprivation was performed in this protocol. In the FR1 phase (10 days), the aim was to evaluate the number of responses on the active nose-poke, the 20% ethanol (*v/v*) intake and the motivation to drink. After each session, the alcohol that remained in the receptacle was collected and measured with a micropipette. To achieve this goal, the number of effective responses and ethanol consumption (μL) were measured under a fixed ratio 1 (FR1) for 10 consecutive daily sessions. The PR phase (1 day) was completed to establish the breaking point for each animal (the maximum number of nose-pokes each animal is able to perform to earn one reinforcement). The response requirement to achieve reinforcements escalated according to the following series: 1-2-3-5-12-18-27-40-60-90-135-200-300-450-675-1000. To evaluate motivation toward ethanol consumption, the breaking point was calculated for each animal as the maximum number of consecutive responses it performed to achieve one reinforcement according to the previous scale. For example, if an animal activated the nose-poke a total of 108 times, this meant that it was able to respond a maximum of 40 times consecutively for one reinforcement. Therefore, the breaking point value for this animal would be 40. All the sessions lasted one hour, except the PR session, which lasted two hours.

#### 2.3.10. Tissue Sampling and Determination of Striatal IL-6

Animals were sacrificed by cervical dislocation and then decapitated to collect blood from the neck in tubes coated with heparin. Blood samples were kept on ice, and plasma was separated from whole blood by centrifugation (5 min, 5000 G) and transferred to sterile 0.2 mL microcentrifuge tubes. To obtain striatum samples, brains were removed immediately after decapitation and dissected following the procedure described by Heffner et al. [50]. Tissue samples were stored at −80 °C until IL-6 determination. To determine striatal IL-6 concentration, we used a Mouse IL-6 ELISA Kit obtained from Abcam (Ref: Ab100712) following the manufacturer’s instructions. Before running the kit, striatum samples were first homogenized and prepared following the procedure described by Ferrer-Pérez et al. [37], and protein levels were determined by the Bradford assay from ThermoFisher (Ref: 23227).

### 2.4. Statistical Analyses 

The body weight data were analyzed using a two-way ANOVA with a between-subjects variable (control and VSD) and a within-subjects variable (weeks with 9 levels). 

For the analysis of the biochemical data, a two-way ANOVA with the same two between-subjects variables (control and VSD) and a within-subjects variable, sessions, with 2 levels (30 min and 4 h), was performed to analyze the data of corticosterone levels. 

The establishment of CPP was determined using a two-way ANOVA with a between-subjects variable, VSD, with two levels (control and VSD) and a within-subjects variable, Days, with two levels (pre-c and post-c). 

To analyze DID and acquisition of ethanol SA, a two-way ANOVA was performed with a between-subjects variable, VSD, with two levels (control and VSD) and a within-subjects variable, days, with four or ten levels for DID and FR1, respectively. The effects of VSD on breaking point values and ethanol consumption during PR were analyzed by a one-way ANOVA with a between-subjects variable, VSD.

Similarly, all behavioral data were analyzed using a one-way ANOVA with a between-subjects variable (control and VSD), except for the data of the time spent in the corners, for which a two-way ANOVA was performed with a between-subjects variable, VSD, with two levels (control and VSD) and a within-subjects variable, sessions, with two levels (object vs. social). Striatal Il-6 levels were analyzed using a one-way ANOVA with a between-subjects variable (control and VSD, post-VSD and post-CPP). 

Data are presented as mean ± SEM, and a *p*-value < 0.05 was considered statistically significant. Analyses were performed using SPSS v26. In all cases, post hoc comparisons were performed with Bonferroni tests.

## 3. Results

### 3.1. VSD Female Showed Increased Corticosterone Levels after Vicarious Experience

The ANOVA of the circulating corticosterone levels 30 min after the 1st and the 4th VSD showed an effect of the variable stress (F (1,14) = 5035; *p* < 0.042). After 30 min of the 1st and the 4th VSD encounters, females presented higher corticosterone levels than controls (*p* < 0.05). The ANOVA after 4 h of the 1st and the 4th VSD did not show any effect, meaning that the levels of corticosterone had returned to normal. The increase in corticosterone levels observed in VSD females seems to be short-lasting (see Figure 2). 

### 3.2. VSD Did Not Affect Bodyweight

The ANOVA of the body weight (see Figure 3) only showed an effect of the variable weeks (F (8304) = 76,408; *p* < 0.001). All females, irrelevant of the stress exposure, increased their body weight across the experimental procedure. Weight in the 1st and 2nd weeks was significantly lower than those of the six following weeks (ps < 0.001). Equally, bodyweight during 3rd, 4th, 5th and 6th weeks were lower than weeks 7th, 8th and 9th (ps < 0.001). 

### 3.3. VSD Females Did Not Show Social Avoidance or Despair Behaviors

The ANOVA of the SIT data performed after the last SD (see Figure 4) did not show a significant effect of the variable stress (F (1,32) = 0.675; *p* < 0.418). The analyses of the SIT, calculated by considering the time spent by an experimental mouse in the interaction zone when a social target is present divided by the time it spends in the interaction zone when the target is absent, did not show any significant effect. Control and VSD female mice spent more time in the social area during the social session, presenting a ratio superior to 1. However, the data of the time spent in the corners (F (1,30) = 7.123; *p* < 0.012) revealed that VSD females spent more time in corners during the object session with respect to non-stressed females (*p* < 0.05), significantly decreasing time in corners during the social session (*p* < 0.001). 

In the TST, the ANOVA for the latency or the time spent in immobility did not show any significant effects (see Figure 5).

### 3.4. VSD Females Showed Increased Anxiety and Anhedonia

The data of the EPM test are presented in Table 1. The ANOVA of the time spent in the open arms (F (1,26) = 4.502; *p* < 0.044) and the percentage of time spent in the open arms (F (1,26) = 5.807; *p* < 0.023) revealed a significant effect of the variable stress. Post hoc analyses showed that VSD females spent less time and a lower percentage of time in the open arms than control non-stressed females (*p* < 0.05 in both cases).

In the ST, the ANOVA for total grooming showed a significant effect of the variable stress (F (1,30) = 5.805; *p* < 0.023), showing a lower percentage of self-grooming time in VSD females than in control mice (*p* < 0.05) (see Figure 6). The ANOVA for back grooming also showed a significant effect of the stress variable (F (1,28) = 8.269; *p* < 0.008), with VSD females displaying less back grooming than controls (*p* < 0.01). However, the ANOVA for leg grooming did not reveal any significant effect.

### 3.5. VSD Females Showed Increased Conditioned Rewarding Effects of Cocaine

The ANOVA of the conditioning score [F (1,26) = 12.004, *p* < 0.002] showed higher scores in VSD groups with respect to controls (*p* < 0.01) (see Figure 7). The ANOVA of the time spent in the drug-paired compartment also showed a significant effect of the interaction Day x Stress [F (1,26) = 10.395, *p* < 0.003]. Only VSD females increased the time in the cocaine-paired compartment during the Post-C test (*p* < 0.001).

### 3.6. VSD Females Showed Increased Ethanol Intake

The ANOVA for the g/kg of ethanol intake during the DID (Figure 8) showed an effect of the variable Days [F (3111) = 16.991; *p* < 0.001] and Treatment [F (1,37) = 17.252; *p* < 0.001]. During the 4th day, all females ingested more ethanol than any of the three previous days (p’s < 0.001 in all cases). VSD females ingested more ethanol with respect to non-stressed females on all 4 days of the procedure (*p* < 0.001).

The ANOVA for the ethanol consumption (g/kg) during the FR1 schedule revealed a significant effect of the variable Stress [F (1,24) = 6.928; *p* < 0.015]. Female mice exposed to VSD exhibited increased ethanol consumption compared to controls (*p* < 0.01). With respect to the number of effective responses during FR1, the ANOVA did not reveal any significant effect.

During the PR, the ANOVA for the breaking point values of ethanol SA did not reveal any significant effect. The ANOVA did not show any differences for ethanol consumption during PR.

### 3.7. VSD Did Not Increase Striatal IL-6 Levels in Female Mice

Regarding striatal IL-6 levels (Figure 9), the ANOVA did not reveal significant differences between control and VSD females, neither after the last VSD nor after CPP (F (3,26) = 0.154, *p* < 0.926).

## 4. Discussion

It is well known that social conflicts in human beings increase the appearance of stress-related pathologies, and the SD model allows for an accurate assessment of the consequences of these social interactions [51]. Humans and rodents are highly social animals and share the adaptive capacity of being influenced by the emotional state of fellow conspecifics [52]. However, as occurs with other adaptive behaviors, if these emotional and physiological empathic responses are intense or prolonged, they can induce similar pathologies to those observed in stressed subjects [53]. The VSD model is based on the presence of behavioral and physiological consequences of social stress in subjects not physically exposed to stress. This model allows us to evaluate female responses while also considering that, among women, psychological aggression commonly involves fewer physical interactions [54]. It is important to highlight that women are more vulnerable to the effects of social stress on drug use [11,12,13]. The results of this study have shown that this model proved to induce a wide range of hormonal and behavioral endophenotypes characteristic of SD effects typically described in defeated male mice. 

### 4.1. VSD Induced an Increase in Corticosterone Levels 

When male mice lose agonistic encounters, they present hypothalamic–pituitary adrenal axis hyperactivity, which increases corticosterone levels [51]. As with defeated males, females exposed to VSD showed a significant increase in corticosterone levels after the 1st and the 4th SD of short duration. The increase in the corticosterone levels of stressed females was 50% after the 1st SD and 85% after the 4th. Although the increment after the 1st SD was lower in females than that observed in stressed male mice in previous studies (which increased more than 100% compared to non-stressed male mice), the observed increase after the 4th SD was higher in VSD females than in males (which showed a 50% increment) [55]. Therefore, we can conclude that VSD induces a potent increment in corticosterone levels that can be comparable to that observed in physically defeated male mice.

We did not expect to find any changes in the body weight of females exposed to VSD, as no decrements are usually observed in defeated male mice exposed to four SD encounters. We must take into account that our protocol is short (25 min) and intermittent (each 72 h). The lack of corticosterone increments 4 h after the end of the VSD indicates the normalization of the physiological reaction of the females and points to the fact that no further physiological alterations should happen.

### 4.2. VSD Increased the Conditioned Rewarding Effects of Cocaine

Women present a higher vulnerability to cocaine addiction in every phase of the addiction cycle, having a more intense response to stress [56]. For instance, in response to a stressful stimulus, women experience negative emotions more intensely, show a higher craving for cocaine and relapse more often [57,58,59]. However, although it has been well established that socially defeated male mice develop a long-lasting increase in the conditioned rewarding effects of cocaine, developing a preference in the CPP with a non-effective dose of cocaine [32,35,40], no studies to date have been conducted in socially stressed female mice.

Our results showed, for the first time, that VSD induced similar effects in stressed females as SD did in male mice regarding the increased rewarding effects of cocaine. Three weeks after the last VSD, stressed females developed a preference for a non-effective cocaine dose (1.5 mg/kg). This 1.5 mg/kg dose of cocaine was employed, as we observed in previous studies that this dose was non-effective, given that female mice require higher cocaine doses than males to develop a preference. Vicariously defeated females increased the time spent in the cocaine-paired compartment after conditioning by 94 s, with this increase being similar to or even higher than that observed in physically defeated males [35,37,60].

Our results extended those of previous studies using aggressive lactating dams to induce social stress in female rats. Ten days after the last of four SD encounters, defeated female rats presented an increase in self-administered cocaine compared to non-stressed controls [61]. In this study, stressed females administered a higher number of infusions and binged for significantly longer than non-stressed females. Similar results were obtained using chronic SD stress, where lactating resident females were confronted for 21 days with the experimental females, which remained housed but protected by the resident female during this period [62]. 

Although these two previous studies, along with ours, support the idea that socially stressed females developed long-lasting, enhanced sensitivity to the rewarding effects of cocaine, a recent study using social isolation during adolescence showed an increased preference for cocaine only in male mice [63]. Therefore, specific studies on adolescent stressed females are needed.

### 4.3. VSD Increased Ethanol Intake in Stressed Females

To date, numerous reports have confirmed that SD produces a long-lasting increase in ethanol intake using the oral ethanol SA paradigm [38,41,64]. To our knowledge, this is the first report showing that VSD female mice presented a long-lasting increase in ethanol SA. Stressed females presented, three weeks after the last SD, an increase in the number of effective responses and more ethanol intake than non-stressed controls. However, no increase in the motivation to obtain ethanol was observed. In contrast with defeated male mice, VSD females showed similar BP to control females. Several reasons could explain this lack of an effect. We can consider that the distress induced by the VSD could be less intense than that experienced by physically defeated mice. Alternatively, the high number of active responses made by control females during the progressive ratio (BP 25) in comparison with control male mice (BP less than 15) could mask an increase in motivation for ethanol.

Since caloric intake and satiety are some components of feeding behavior that are also important modulators of oral alcohol SA, we have to point out that the protocol of alcohol SA employed in this study did not require food deprivation. We have previously validated this oral SA procedure in male mice [65]. In the present study, we have confirmed that this SA proved to be sensitive to detecting long-lasting social stress effects. 

### 4.4. VSD Induce Anxiety-and Anhedonic-like Behaviors

It is well known that SD induces a clear anxiogenic effect [35,37,66,67]. Typically, defeated adult male mice showed a 50% decrease in the time and the percentage of time spent in the open arms of the EPM compared to control non-stressed male mice. We have obtained similar results in VSD females, which showed a 43% decrease in the time and percentage of time spent in the open arms. In line with our results, several chronic stress models used in female rats, such as chronic mild stress [68], social instability [69] or physical aggression [21], report increased anxiety-like behaviors in the EPM or the open field.

Depressive-like states in humans are characterized by symptoms of apathy and anhedonia, which can be assessed in mice by measuring spontaneous grooming behavior with the ST [46,70]. Total and back grooming time in the ST is an index of motivational and self-care behavior [70], this form of motivational behavior being parallel with some symptoms of depression, such as apathetic behavior [71]. It is known that rats subjected to chronic, unpredictable, mild stress for more than 40 days showed an increased latency to initiate and decreased time of grooming behavior [72].

Interestingly, social contagion shows that cohabitation with a conspecific in chronic pain induces, besides hypernociception and antinociception, a depressive-like effect in the sucrose ST in non-affected mice [73]. Our results confirmed that VSD females showed apathy represented by lower total and back grooming than control non-stressed females.

Defeated mice can also present depressive-like symptoms, such as social withdrawal, anhedonia, or behavioral despair [74,75,76]. The TST is a well-validated method for the evaluation of the antidepressant efficacy of drugs and can be used to evaluate the effects of environmental manipulations such as social stress [47,77]. In the TST, which puts mice in an inescapable but moderately stressful situation, immobility is considered to be the lack of escape-related behavior. Although chronic SD stress is known to cause increased immobility in the TST [78,79], with our protocol, in which male mice experience only 4 SD separated by 72 h, we did not observe an increase in immobility time in the TST in defeated adult male mice [80]. Most studies in male rodents report that a short-period paradigm of SD is more likely to result in the phenotype of anxiety [81] since other reports have indicated that 20 days of social stress is required to develop depression [82]. Therefore, it was expected that VSD females would not show changes in this test. The longer exposure to SD, as observed in the Iñiguez study [31], could explain the increased time in immobility observed in females. Interestingly and converging with our results, using a chronic pain cohabitation procedure, Baptista-de-Souza and co-workers [83] did not observe significant results for the TST.

The SIT is designed to measure social preference or social avoidance behaviors and operates on the premise that rodents prefer social to non-social stimuli. This test allows us to examine social investigation in a controlled setting. In most studies, either after a chronic or intermittent SD protocol, defeated male mice showed a SIT ratio lower than 1, meaning a decrease in social contacts [19,40]. However, VSD females showed a higher SIT ratio than 1 and even spent less time in the corners during the subject test, meaning that they tended to socialize with non-threatening males. Our results agree with previously reported results using different protocols to stress female rodents [24,84,85,86].

There are several possible explanations for this low percentage of susceptible females to social avoidance. The increased social preference for familiar or non-threatening social stimuli in defeated females could be due to the reported “tend and befriend” phenomenon observed in some stressed human and rodent females [87]. Another possible explanation is the confounding of sexual motivation. Interestingly, when defeated females were allowed, many of them moved to the male’s compartment, suggesting that defeated females do not find the 24 h psychosensory exposure period aversive [24].

Although there is consensus regarding the increased neuroinflammatory response after exposure to SD, the vicarious experience in rodent males does not seem to induce a substantial activation of the immune system [88,89]. Fewer data are available for defeated female rodents, showing that similarly to males, physically defeated females also exhibit increased neuroinflammatory markers [24,25]. Our results agree with that reported in VSD males since we did not observe increases in striatal IL-6 levels, either after the last VSD or after cocaine-induced CPP. Due to the complexity of the neuroinflammatory response, more studies evaluating other inflammatory markers are needed.

### 4.5. Limitations and Future Perspectives

As with most of the recent studies using female mice, we did not determine the phase of the estrous cycle [31]. In addition, in the study of Holly and co-workers [61], in which the estrous cycle was evaluated, the authors did not find any cycle effect, and therefore, all the cocaine SA data were collapsed together. Despite these data, we cannot discard the fact that fluctuations in female hormones could affect the results observed, taking into account that females were housed with a male mouse for four periods of 24 h.

More studies are needed to dissect the neurobiological and behavioral peculiarities of the female response to social stress, highlighting the study of the neuroinflammatory response to VSD.

The higher rate of psychopathologies in women and the sex differences in the contributions of stress to drug misuse indicated the need for specific studies on the female response to social stressors. Rodent models of social stress validated for females allow us to investigate the particular behavioral and neurochemical consequences of stress in females, as well as the specific mechanisms of vulnerability to increased depression, anxiety or drug use disorders. To date, only a small percentage of experiments have been performed in female rodents; however, it is women who have the highest prevalence of disorders induced by psychosocial stress [11,12,13]. Although the stressors are different and VSD cannot be compared to SD in males, we have obtained comparable behavioral phenotypes in females to those classically observed in defeated male mice.

## 5. Conclusions

Our results indicate that female mice vicariously experience a state of distress through the social observation of others suffering from adverse events. Most studies point to visual cues being critical in the vicarious experience [31,90]. Similar phenotypes in vicariously defeated males to those observed in defeated male mice have also been reported [28]. Interestingly, in these studies, gene expression dysregulation in the ventral tegmental area [28] and the nucleus accumbens [91] was comparable between witnesses and intruders, highlighting the similar nature of the response to social stress with both procedures.

Most of the procedures usually employed to induce social stress implement artificial stressors over extended periods of time that are unnatural to the animal (i.e., restrain, foot-shock, etc.) or involve physical attack (SD stress). These experimental approaches do not represent the more common types of stress in humans, which are social and emotional non-physically mediated stressors. The female VDS model presented in this study induces most of the critical behavioral features observed in socially defeated stressed male mice without the confounding variable of physical confrontation. This model could help to disectionate the neurobiological basis of psychopathology induced by social stressors and to support the development of new strategies for the prevention and treatment of drug misuse, especially in women.

## Figures and Tables

**Figure 1 biomedicines-11-00502-f001:**
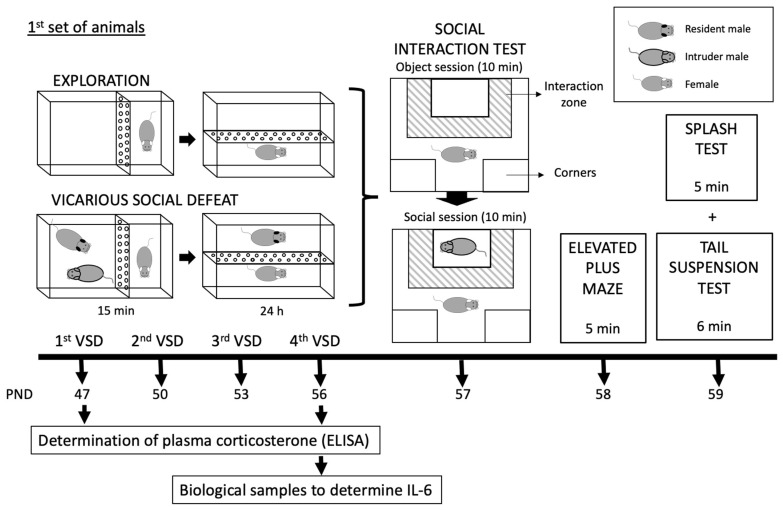
Experimental design and experimental groups of two mice sets.

**Figure 2 biomedicines-11-00502-f002:**
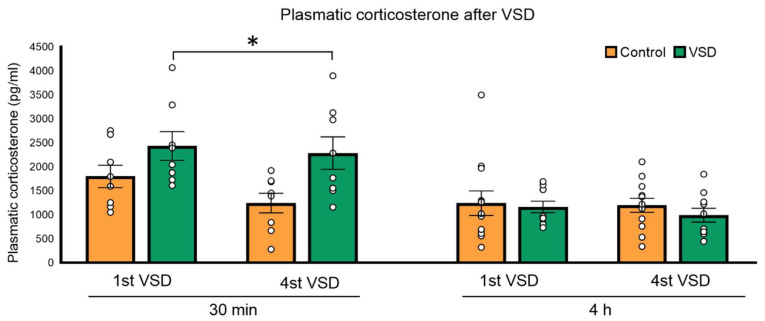
Vicarious Social Defeat (VSD) increased plasmatic corticosterone levels in female mice. Bars represent the mean of the plasmatic corticosterone levels (in pg/mL) 30 min (control, n = 8; VSD, n = 8) and 4 h (control, n = 8; VSD, n = 8) after the 1st and 4th social defeat (SD) in control and vicariously defeated females and the vertical lines ± SEM. Stress effect (F (1,14) = 5035; *p* < 0.042); * *p* < 0.05, significant difference compared to the control group.

**Figure 3 biomedicines-11-00502-f003:**
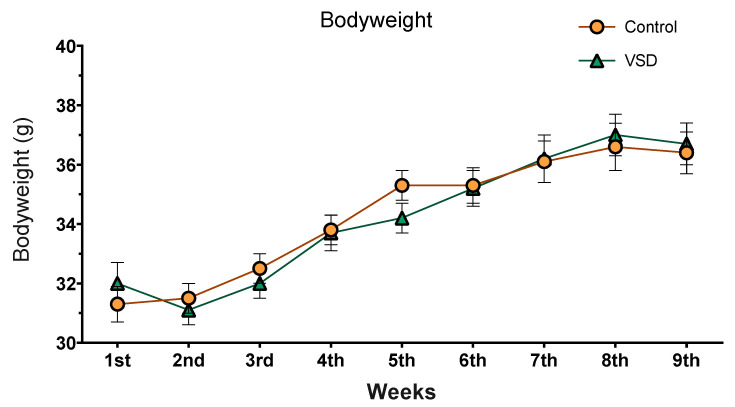
Body weight of mice over 9 weeks. Data are presented as mean (±SEM) amount of body weight (control group, n = 20; and VSD group, n = 20).

**Figure 4 biomedicines-11-00502-f004:**
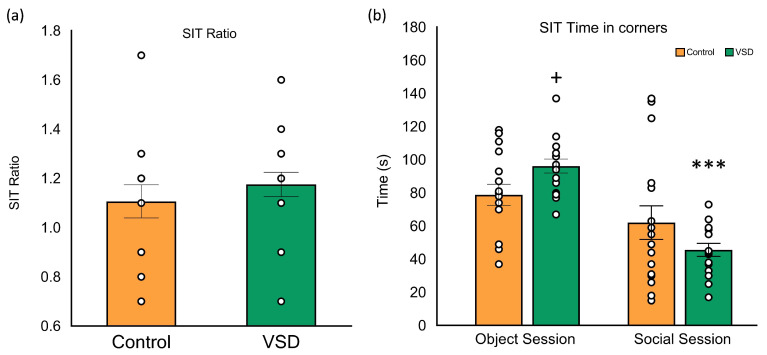
Vicarious Social Defeat VSD did not alter the Social Interaction Test (SIT) ratio in female mice. The bars represent the ratio of the SIT (**a**) and the mean of the time spent in corners during the object and social sessions (**b**), and the vertical lines, ±SEM. Values > 1 indicate preference for social interaction, and <1 indicates social avoidance. Female mice were divided into control (n = 16) and VSD (n = 16). Interaction stress x sessions: (F (1,30) = 7.123; *p* < 0.012), Bonferroni post hoc test + *p* < 0.05, significant difference compared to the control group; *** *p* < 0.001, significant difference compared to the object session.

**Figure 5 biomedicines-11-00502-f005:**
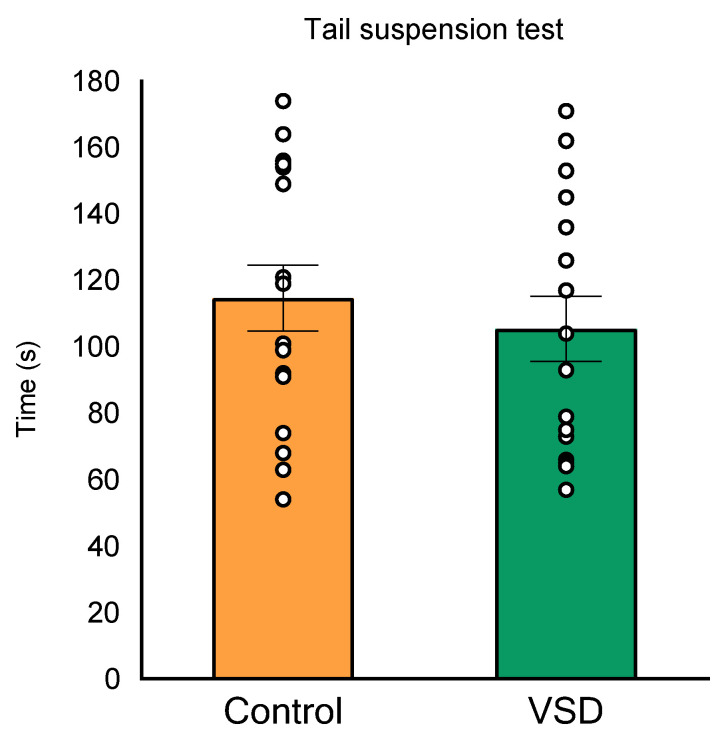
Vicarious Social Defeat (VSD) did not increase time in immobility in the tail suspension test (TST). Bars depict the mean ± SEM of the time (seconds) during which the mice remained immobile in the TST (control group, n = 16; VSD group, n = 16).

**Figure 6 biomedicines-11-00502-f006:**
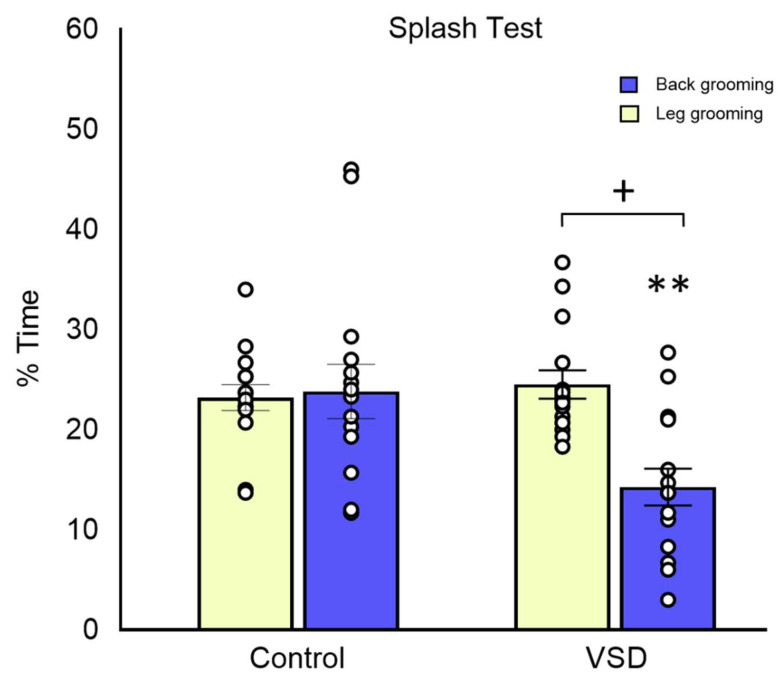
Vicarious Social Defeat (VSD) in female mice increases anhedonia in the splash test (ST). The bars depict the percentage of time in leg and back grooming in the ST (Control n = 15; VSD n = 15). Stress effect in total grooming [F (1,30) = 5.805; *p* < 0.023], + *p* < 0.05 significant differences with respect to controls. Stress effect in back grooming [F (1,28) = 8.269; *p* < 0.008], ** *p*< 0.01 significant differences with respect to controls.

**Figure 7 biomedicines-11-00502-f007:**
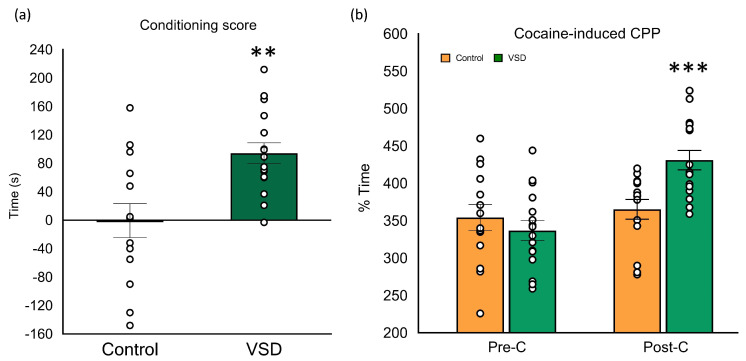
Vicarious Social Defeat (VSD) female mice showed preference in cocaine-induced Conditioned Place Preference (CPP). Female mice were divided into Control (n = 14) and VSD (n = 14). (**a**) The bars represent the conditioning score (difference in seconds between the time spent in the drug-paired compartment after the conditioning sessions and the time spent in the same compartment during Pre-C); or (**b**) the time (in seconds) spent in the drug-paired compartment before conditioning sessions in the pre-conditioning test (Pre-C) (light gray bars) and after conditioning sessions in the post-conditioning test (Post-C) (dark gray bars), during which CPP was induced with 1,5 mg/kg of cocaine. Stress effect in the conditioning score [F (1,26) = 12.004, *p* < 0.002], ** *p* < 0.01 significant difference with respect to the control group. Interaction Day x Stress in the time spent in the drug-paired compartment [F (1,26) = 10.395, *p* < 0.003], *** *p* < 0.001 significant difference in the time spent in the drug-paired compartment vs. Pre-C session.

**Figure 8 biomedicines-11-00502-f008:**
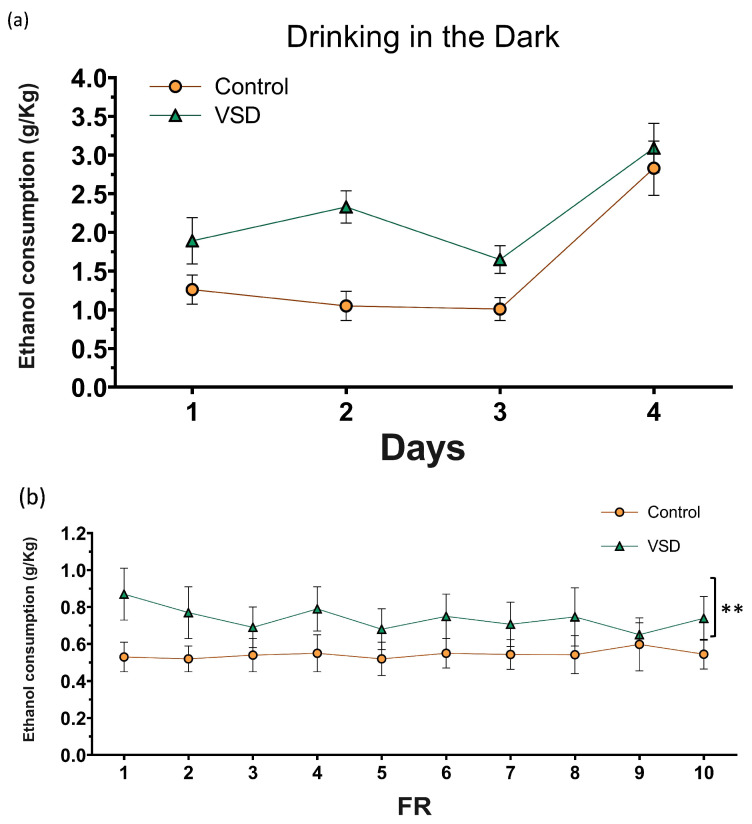
(**a**) Effect of Vicarious Social Defeat (VSD) on ethanol intake during Drinking in the Dark (DID). The dots represent means and the vertical lines ± SEM of the g/kg of ethanol at 20% consumed. (**b**) Oral ethanol Self-Administration (SA). The dots represent means and the vertical lines ± SEM of the volume of 20% ethanol consumption during FR1 (in g/kg) and the stress effect (F (1,24) = 6.928; *p* < 0.015). ** *p* < 0.01, significant difference in ethanol consumption with respect to the control group. (**c**) The number of active responses. (**d**) The columns represent means and the vertical lines ± SEM of breaking point values and the volume of 20% ethanol consumption (in g/kg) obtained during PR (control group, n = 14; VSD group, n = 12).

**Figure 9 biomedicines-11-00502-f009:**
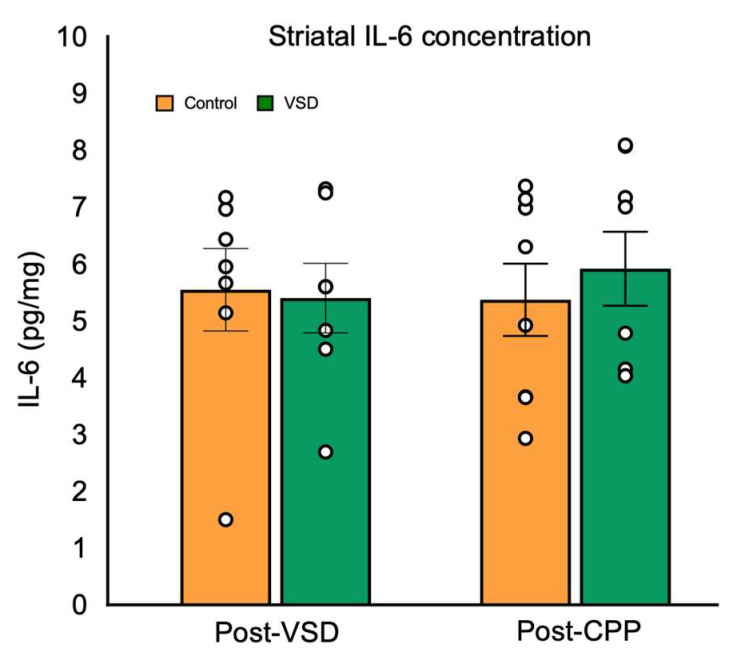
Vicarious Social Defeat (VSD) does not affect IL-6 levels in the striatum. Female mice were divided into control (n = 7 after VSD and n= 8 after CPP) and VSD (n = 7 after VSD and n= 8 after CPP). The bars represent the mean values ± SEM (pg/mg).

**Table 1 biomedicines-11-00502-t001:** VDS increases anxiety in female mice. * *p* < 0.05 significant difference compared to the control group.

	Control	VSD
**Time OA**	85 ± 13	49 ± 12 *
**% Time OA**	43 ± 5	25 ± 6 *
**Time center**	112 ± 12	115 ± 6
**Entries OA**	22 ± 3	17 ± 2
**% Entries OA**	51 ± 6	39 ± 4
**Total Entries**	43 ± 3	42 ± 3

## Data Availability

The data presented in this study are available on request from the corresponding author.

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
