# Peer review of "Vicarious Social Defeat Increases Conditioned Rewarding Effects of Cocaine and Ethanol Intake in Female Mice"

_biomedicines, 2023, doi:10.3390/biomedicines11020502_

Round 1

Reviewer 1 Report

This is a very interesting paper about the effect of vicarious social defeat on the effects of cocaine and ethanol intake in female mice. The paper is well-written and of interest for the readers. However, several minor changes should be amde before considering it for publication.

Abstract.

1- I recommend to reduce the introduction section of the abstract. I just suggest to mention that stress is a factor influencing the development of mood and substance use disorders, and the effects of social defeat in female mice models have been poorly investigated.

2- Female mice experience an increased preference for a dose of cocaine and ethanol intake. How are the authors explaining these findings in the context of humans? Can they build a hypothesis for further studies?

Introduction.

1- The authors start the paper reporting that stress is a critical factor in the development of mood disorders, depression or anxiety. I recommend to expand this paragraph by adding some evidence about which hormones or hormonal pathways have been implication in that relationship (stress-depression).

2- The last paragraph of the introduction is focused on the main aims of the paper. 

I recommend to describe them into a separate subsection (1.1. Aims).

Material and methods.

1-The subsection called "2.1. Animals" can be renamed as Study design. Figure 1 is excellent to better understand how the experients were done. 

2- The authors used the determination of plasma corticosterone for the assessment of stress hormones. Perhaps, it can be explained in detail in the introduction section. 

3- PLease, expand the explanation about the Splash test. Why is it necessary in terms of the aims of the present study?

Results.

1- The results section is really good. The authors have cemphasized the main results in the different subsections of the results, and it makes the results more clear.

Conclusions

1- The conclusions section is to long. The authors are describing once again the main results, and they are discussing them according to the literature (and presenting cites/references).

I recommend to summarize this section. The last paragraph which is mainly about future perspectives, can be moved to a new subsection of the discussion section called "Future perspectives". Future studies considering these results in human populations are welcome.

Author Response

Abstract.

  • I recommend to reduce the introduction section of the abstract. I just suggest to mention that stress is a factor influencing the development of mood and substance use disorders, and the effects of social defeat in female mice models have been poorly investigated.

Following the reviewer’s suggestion, the abstract has been modified as follows:

Stress is a critical factor in the development of mood and drug use disorders. The social defeat model is not appropriate for female rodents due to their low level of aggression. Therefore, a robust female model of social stress needs to be developed and validated.

2- Female mice experience an increased preference for a dose of cocaine and ethanol intake. How are the authors explaining these findings in the context of humans? Can they build a hypothesis for further studies?

We have added the following hypothesis at the end of the abstract:

The fact that VSD in females induced a comparable behavioral phenotype to that observed in physically defeated males could indicate a relation with the higher rate of psychopathologies observed in women.

Introduction.

1- The authors start the paper reporting that stress is a critical factor in the development of mood disorders, depression or anxiety. I recommend to expand this paragraph by adding some evidence about which hormones or hormonal pathways have been implication in that relationship (stress-depression).

Following the reviewer’s suggestion, we have added the following information to the I     ntroduction section:

For instance, the neurotransmitter serotonin has been closely associated with the appearance of psychiatric disorders such as depression and anxiety (Narayanan et al., 2011), but it is also deeply affected by the experience of stress (Harvey et al., 2004). Stress-induced 5-HT activity occurs in areas such as the ventral striatum (Amato et al., 2007), the hippocampus (Keeney et al., 2006), or the cerebellum (Azevedo et al., 2020). In addition, social stress increased the kynurenine pathway of tryptophan metabolism, with numerous studies linking the kynurenine pathway with neuroinflammation and depression (Wang et al., 2018; Laumet et al., 2017).

Narayanan, V., Heiming, R.S., Jansen, F., Lesting, J., Sachser, N., Pape, H.C., Seidenbecher, T., 2011. Social defeat: impact on fear extinction and amygdala- prefrontal cortical theta synchrony in 5-HTT deficient mice. PloS One 6 (7), e22600. https://doi.org/10.1371/journal.pone.0022600.

Harvey, B.H., Naciti, C., Brand, L., Stein, D.J., 2004. Serotonin and stress: protective or malevolent actions in the biobehavioral response to repeated trauma? Ann. N. Y. Acad. Sci. 1032, 267–272. https://doi.org/10.1196/annals.1314.035.

Amato, J.L., Bankson, M.G., Yamamoto, B.K., 2007. Prior exposure to chronic stress and MDMA potentiates mesoaccumbens dopamine release mediated by the 5-HT(1B) receptor. Neuropsychopharmacology: official publication of the American College of Neuropsychopharmacology 32 (4), 946–954. https://doi.org/10.1038/sj. npp.1301174.

Keeney, A., Jessop, D.S., Harbuz, M.S., Marsden, C.A., Hogg, S., Blackburn-Munro, R.E., 2006. Differential effects of acute and chronic social defeat stress on hypothalamic- pituitary-adrenal axis function and hippocampal serotonin release in miceJ. Neuroendocrinol. 18 (5), 330–338. https://doi.org/10.1111/j.1365- 2826.2006.01422.x.

Azevedo, H., Ferreira, M., Mascarello, A., Osten, P., Werneck Guimara ̃es, C.R., 2020. The serotonergic and alpha-1 adrenergic receptor modulator ACH-000029 ameliorates anxiety-like behavior in a post-traumatic stress disorder model. Neuropharmacology 164, 107912. https://doi.org/10.1016/j.neuropharm.2019.107912.

Wang, J., Hodes, G.E., Zhang, H., Zhang, S., Zhao, W., Golden, S.A., Bi, W., Menard, C., Kana, V., Leboeuf, M., Xie, M., Bregman, D., Pfau, M.L., Flanigan, M.E., Esteban- Fern ́andez, A., Yemul, S., Sharma, A., Ho, L., Dixon, R., Merad, M., Pasinetti, G.M., 2018. Epigenetic modulation of inflammation and synaptic plasticity promotes resilience against stress in mice. Nat. Commun. 9 (1), 477. https://doi.org/10.1038/ s41467-017-02794-5.

Laumet, G., Zhou, W., Dantzer, R., Edralin, J.D., Huo, X., Budac, D.P., O’Connor, J.C., Lee, A.W., Heijnen, C.J., Kavelaars, A., 2017. Upregulation of neuronal kynurenine 3-monooxygenase mediates depression-like behavior in a mouse model of neuropathic pain. Brain Behav. Immun. 66, 94–102. https://doi.org/10.1016/j. bbi.2017.07.008.

2- The last paragraph of the introduction is focused on the main aims of the paper. 

I recommend to describe them into a separate subsection (1.1. Aims).

We have generated a new subheading 1.1. Aims.

Material and methods.

1-The subsection called "2.1. Animals" can be renamed as Study design. Figure 1 is excellent to better understand how the experients were done. 

We have renamed subsection 2.1. as Study design and merged with the subsection 2.3. Experimental design.

2- The authors used the determination of plasma corticosterone for the assessment of stress hormones. Perhaps, it can be explained in detail in the introduction section. 

We have added this information to the Introduction section in subsection 1.1, as follows:

The aim of the present study was to unravel the effects of VSD on the cocaine conditioned rewarding effects and ethanol intake in female mice, confirming that it induces an increase in corticosterone response.

3- Please, expand the explanation about the Splash test. Why is it necessary in terms of the aims of the present study?

One of our aims was to study the appearance of depressive-like behaviors in females exposed to VSD. In addition to anhedonia, behavioral endpoints for measuring the anxio-depressive phenotype in rodents include social aversion (measured in our study with the social interaction test), despair (measured in our study with the tail suspension test), anxiety (measured in our study with the elevated plus maze), and apathy, measured with the splash test. Various tests have been employed in order to measure these different aspects of depression

Apathy has been defined as a deficit in goal-directed behavior (Levy and Dubois 2006). The splash test consists in splashing a sucrose solution over the coat of the animal and measuring the grooming behavior that has been induced. A decrease in grooming in the back of the animals can be considered a symptom of apathy.

The following information has been added to the subsection 2.3.5. Splash test

A decrease in grooming in the back of the animals can be considered a symptom of apathy, one of the behavioral endpoints for measuring the anxio-depressive phenotype in rodents (Planchez et al., 2019).

Results.

1- The results section is really good. The authors have cemphasized the main results in the different subsections of the results, and it makes the results more clear.

We thank the reviewer for his/her positive comments

Conclusions

1- The conclusions section is to long. The authors are describing once again the main results, and they are discussing them according to the literature (and presenting cites/references).

I recommend to summarize this section. The last paragraph which is mainly about future perspectives, can be moved to a new subsection of the discussion section called "Future perspectives". Future studies considering these results in human populations are welcome.

We have shortened the conclusion section and generated a new section entitled “Limitations and Future perspectives.”

Reviewer 2 Report

The MS "Vicarious social defeat increases conditioned rewarding effects of cocaine and ethanol intake in female mice" is well written and clear. 

The methodology as well as the study design are informative and well described. The study has all the controls needed and is thoroughly conducted. 

Minor:

Introduction: I would suggest to clearly state the hypothesis 

The long section about females is interesting but seems too long and it's not really clear and confusing. You had a clear explanation in the discussion section "Although numerous studies have undoubtedly confirmed the increased rewarding effects of cocaine and ethanol after exposure to SD on male mice, no studies have previously evaluated these effects on socially stressed female mice. In this study, we have employed a VSD model in which females were not physically attacked but witnessed a standard social defeat between two males and continued confined with the resident aggressive male for a period of 24 h." I think this should be included in the introduction. 

Results:

It would help to add color to the graphs and curve. 

Figure 3, 5 I would suggest to add the "n" in the legend

Figure 8 b the ** should be aligned with the curves. 

Discussion:

I would suggest to reduce the discussion section since most of the discussion focusing on summarizing results.

I would suggest to discuss more the results and put the emphasis on why the results are interesting and the perspectives of the work. 

Also I would suggest to discuss the limitations of this study. 

Major: 

I highly suggest to analyze the inflammatory response after the stress using ELISA or western blot or flow cytometry, .....

Author Response

Minor:

Introduction: I would suggest to clearly state the hypothesis 

 We have clarified the hypothesis of the study, as follows:

We hypothesized that females exposed to VSD will show a comparable phenotype to      that of physically defeated male mice, with increased corticosterone levels, higher levels of anxiety and depressive-like behaviors. We also hypothesized that the effect of VSD will last in time, as vicariously defeated females will develop preference for a non-effective dose of cocaine and will drink more ethanol than non-exposed female controls 3 weeks after the last exposure to VSD.

The long section about females is interesting but seems too long and it's not really clear and confusing. You had a clear explanation in the discussion section "Although numerous studies have undoubtedly confirmed the increased rewarding effects of cocaine and ethanol after exposure to SD on male mice, no studies have previously evaluated these effects on socially stressed female mice. In this study, we have employed a VSD model in which females were not physically attacked but witnessed a standard social defeat between two males and continued confined with the resident aggressive male for a period of 24 h." I think this should be included in the introduction. 

 We have made the changes suggested by the reviewer.

Results:

It would help to add color to the graphs and curve. 

Color has been added to the revised graphs in order to facilitate the interpretation.

Figure 3, 5 I would suggest to add the "n" in the legend

We have added the “n” information in the legend of Figures 3, 5, and 8, following the Reviewer’s suggestion:

Figure 3. Bodyweight of mice over 9 weeks. Data present mean (± SEM) amount of body weight (Control group, n=20; and VSD group, n=20).

Figure 5. VSD did not increase time in immobility in the tail suspension test. Bars depict the mean ± SEM of the time (seconds) during which the mice remained immobile in the tail suspension test (Control group, n=16; and VSD group, n=16).

Figure 8. (a) Effect of VSD on ethanol intake during DID. The dots represent means and the vertical lines ± SEM of the g/kg of ethanol at 20% consumed; (b) Oral EtOH self-administration. The dots represent means and the vertical lines ± SEM of the volume of 20% EtOH consumption during FR1 (in g/kg),  ** p< 0.01 significant difference in ethanol consumption with respect the control group; (c) The number of active responses; (d) The columns represent means and the vertical lines ± SEM of breaking point values and the volume of 20% EtOH consumption (in g/kg) obtained during PR. (Control group, n=14; and VSD group, n=12).

Figure 8 b the ** should be aligned with the curves. 

We have changed the alignment.

Discussion:

I would suggest to reduce the discussion section since most of the discussion focusing on summarizing results.

I would suggest to discuss more the results and put the emphasis on why the results are interesting and the perspectives of the work. 

We have reduced the Discussion section, but considering the other reviewer’s suggestion, we have incorporated comments on references of human studies in the Discussion section to highlight the importance of our results, especially its translational value.

Also I would suggest to discuss the limitations of this study. 

Following to Reviewer’s suggestion, we have included a new section in the Discussion titled “Limitations and future directions.”

 4.5. Limitations and future directions

As with most of the recent studies using female mice, we did not determine the phase of the estrous cycle [23]. In addition, in the study of Holly and co-workers [65], in which the estrous cycle was evaluated, the authors did not find any cycle effect and therefore, all the cocaine self-administration data were collapsed together. Despite this data, we cannot discard the fact that fluctuations in female hormones could affect the results observed, considering that females were housed with a male mouse for four periods of 24 h.

More studies are needed to dissect the neurobiological and behavioral peculiarities of the female response to social stress, highlighting the study of the neuroinflammatory response to VSD.

The higher rate of psychopathologies in women and the sex differences in the contributions of stress to drug misuse indicated the need for specific studies on the female response to social stressors. Rodent models of social stress validated for females allow us to investigate the particular behavioral and neurochemical consequences of stress in females, as well as the specific mechanisms of vulnerability to increased depression, anxiety or drug use disorders. To date, only a small percentage of experiments is performed in female rodents; however, it is women who have the highest prevalence of disorders induced by psychosocial-stress (Peltier et al., 2019; Guinle and Sinha, 2020; Brady et al., 1998). Although the stressors are different and VSD cannot be compared to SD in males, we have obtained comparable behavioral phenotypes in females to those classically observed in defeated male mice.

Major: 

I highly suggest to analyze the inflammatory response after the stress using ELISA or western blot or flow cytometry, .....

We agree with the reviewer that evaluating the inflammatory response to VSD is necessary in this kind of study. We have studied the neuroinflammatory response of SD in male mice mainly through the ELISA analyses of interleuquina 6 and fractalkina (CX3CL1) in the striatum and the hippocampus. We know that SD induces an increased neuroinflammatory response that lasts even after the end of the CPP or the self-administration procedures.

We are currently developing a series of studies in which several cytokines (as Il-6) and chemokines (as Cx3CL1) are being measured in the striatum and the hippocampus of VSD females at different time points. We are now developing these studies and we do not yet have the results. However, preliminary results point to no increases of striatal Il-6 in VSD females after the CPP procedure. We expect to find some increases 24 h after the last VSD experience. 

Reviewer 3 Report

This is an interesting topic. Regarding female humans there is an extensive body of literature that could be included in order to strengthen the theoretical part in the introduction section such as references that discuss mood disorders and drug use disorders:

Guinle, M. I. B., & Sinha, R. (2020). The role of stress, trauma, and negative affect in alcohol misuse and alcohol use disorder in women. Alcohol research: current reviews40(2).

Brady, K. T., Dansky, B. S., Sonne, S. C., & Saladin, M. E. (1998). Posttraumatic stress disorder and cocaine dependence: Order of onset. American Journal on Addictions7(2), 128-135.

Peltier, M. R., Verplaetse, T. L., Mineur, Y. S., Petrakis, I. L., Cosgrove, K. P., Picciotto, M. R., & McKee, S. A. (2019). Sex differences in stress-related alcohol use. Neurobiology of stress10, 100149.

Giannouli, V., & Ivanova, D. (2017). Does comorbid depression and alcoholic dependence influence cognition in Bulgarian womеn?. European Psychiatry41(S1), S473-S473.

Gordon, H. W. (2002). Early environmental stress and biological vulnerability to drug abuse. Psychoneuroendocrinology27(1-2), 115-126.

Ivanova, D., & Giannouli, V. (2017). Lesch type III alcoholism in Bulgarian women: Implications and recommendations for psychotherapy. International Journal of Caring Sciences10(3), 1569-1576.

Shimamoto, A., DeBold, J. F., Holly, E. N., & Miczek, K. A. (2011). Blunted accumbal dopamine response to cocaine following chronic social stress in female rats: exploring a link between depression and drug abuse. Psychopharmacology218(1), 271-279.

In addition to the above, the rationale of the present research must be clearly presented to the reader.

The pictures must be described in detail as the methodology is not clear.

The link between animal findings and human behavior must be described in more detail with the support of relevant literature in the discussion, otherwise the value of this research is not properly presented.

Author Response

This is an interesting topic. Regarding female humans there is an extensive body of literature that could be included in order to strengthen the theoretical part in the introduction section such as references that discuss mood disorders and drug use disorders:

We agree with the reviewer that literature in humans regarding mental disorders and drug use is interesting and will improve the Introduction. We have added a new paragraph taking into account the references suggested by the Reviewer:

Additionally, psychosocial stress factors have long been associated with drug abuse (Gordon, 2002; Koob and Schulkin, 2019), especially in women. Negative affect and stress reactivity are key components in the development of an alcohol use disorder in women (Peltier et al., 2019; Guinle and Sinha, 2020). Moreover, women were more likely than men to have stress precede cocaine use (Brady et al., 1998).

Guinle, M. I. B., & Sinha, R. (2020). The role of stress, trauma, and negative affect in alcohol misuse and alcohol use disorder in women. Alcohol research: current reviews, 40(2).

Brady, K. T., Dansky, B. S., Sonne, S. C., & Saladin, M. E. (1998). Posttraumatic stress disorder and cocaine dependence: Order of onset. American Journal on Addictions, 7(2), 128-135.

Peltier, M. R., Verplaetse, T. L., Mineur, Y. S., Petrakis, I. L., Cosgrove, K. P., Picciotto, M. R., & McKee, S. A. (2019). Sex differences in stress-related alcohol use. Neurobiology of stress, 10, 100149.

Giannouli, V., & Ivanova, D. (2017). Does comorbid depression and alcoholic dependence influence cognition in Bulgarian womеn?. European Psychiatry, 41(S1), S473-S473.

Gordon, H. W. (2002). Early environmental stress and biological vulnerability to drug abuse. Psychoneuroendocrinology, 27(1-2), 115-126.

Ivanova, D., & Giannouli, V. (2017). Lesch type III alcoholism in Bulgarian women: Implications and recommendations for psychotherapy. International Journal of Caring Sciences, 10(3), 1569-1576.

Shimamoto, A., DeBold, J. F., Holly, E. N., & Miczek, K. A. (2011). Blunted accumbal dopamine response to cocaine following chronic social stress in female rats: exploring a link between depression and drug abuse. Psychopharmacology, 218(1), 271-279.

In addition to the above, the rationale of the present research must be clearly presented to the reader.

We have added a new section (1.1. Aims) in which we clarify the rationale of our research and we have included our hypothesis more explicitly:

  • Aims

The aim of the present study was to unravel the effects of VSD on the cocaine conditioned rewarding effects and ethanol intake in female mice, confirming that it induces an increase in corticosterone response. We have repeatedly shown that SD in male rodents induce a long-lasting increase in the conditioned rewarding effects of cocaine evaluated with the conditioned place preference (CPP) paradigm [27, 32,39]. Similarly, SD increased ethanol intake in mice 3 weeks after the last defeat measured with oral ethanol self-administration [35,70]. However, although VSD seems to be a good model to induce behavioral (social avoidance, anhedonia, despair) and physiologic (increased corticosterone and weight change) endophenotypes induced by stress, no studies have been performed to date to characterize the effect of VSD on cocaine or alcohol effects. In a recent study, Newman and co-workers [40] exposed female mice to a 10-day schedule of chronic social defeat stress induced by a resident aggressive female, followed by cohabitation with the aggressor 24h post defeat. Although this study showed defeated female mice drinking more alcohol than controls for 4 weeks after the social defeat, it cannot be considered to be about VSD, since females were physically exposed to an aggressive mouse. We hypothesized that females exposed to VSD will show a comparable phenotype to that of physically defeated male mice, with increased corticosterone levels, higher levels of anxiety and depressive-like behaviors. We also hypothesized that the effect of VSD will last in time, as vicariously defeated females will develop preference for a non-effective dose of cocaine and will drink more ethanol than non-exposed female controls 3 weeks after the last exposure to VSD.

The pictures must be described in detail as the methodology is not clear.

We have revised the description of methodology in the text, including more details for a better explanation of the procedure that has been followed.

The link between animal findings and human behavior must be described in more detail with the support of relevant literature in the discussion, otherwise the value of this research is not properly presented.

Following to Reviewer’s suggestion, we have added relevant literature of human      studies in the Discussion section to support the value of our research.

Reviewer 4 Report

The authors used vicarious social defeat (VSD) to induce behavioral and physiologic endophenotypes induced by stress in female mice and investigated the long-lasting effects of this model on the conditioned rewarding effects of cocaine and ethanol intake. The results indicated that VSD increased anxiety- and depression-like behaviors and delayed increases in cocaine and ethanol intake in female mice, confirming the validity of VSD model to create the response to social stress in female mice. The paper has the potential to contribute to the existing scientific literature on the female model of social stress in animal studies. I only have a few comments to further improve the quality of the authors’ paper. I have outlined these issues below:

1.

1. Please make clear statistical report in each panel. Also, in some of panels (e.g. Figure 4a, 5, table 1), two-way ANOVA analysis was used, please explain the necessity to use two-way ANOVA for comparison between two groups (control vs VSD)
2. There was no description of Figure 7 in Result section.
3. Please discuss why VSD procedure did not cause body weight changed during stress period in VSD group in the discussion (Figure3)
4. Please discuss why there was no increase in motivation to obtain ethanol (figure 8d) in the discussion

In the reviewer’s opinion, the above-mentioned issues need to be addressed by the authors.

Author Response

  1. Please make clear statistical report in each panel. Also, in some of panels (e.g. Figure 4a, 5, table 1), two-way ANOVA analysis was used, please explain the necessity to use two-way ANOVA for comparison between two groups (control vs VSD)

We have improved the statistical report in each panel and thank the Reviewer for his comment, as it has allowed us to detect an error in the Statistical analyses section. The mistake has been corrected:

2.4. Statistical analyses

The bodyweight data were analyzed using a two-way ANOVA with a between-subjects variable (Control and VSD) and a within-subjects variable (weeks with 9 levels).

For the analysis of the biochemical data, a two-way ANOVA with the same two between-subjects variables (Control and VSD) and one within-subjects variable, Sessions with 2 levels (1 and 4) was performed to analyze the data of corticosterone levels at 30 min and 4h.

The establishment of CPP was determined using a two-way ANOVA with a between subjects variable, VSD, with two levels (Control and VSD) and a within-subjects variable, Days, with two levels (Pre-C and Post-C).

To analyze DID and acquisition of ethanol SA, a two-way ANOVA was performed with a between-subjects variable, VSD, with two levels (Control and VSD) and a within-subjects variable, Days, with four or ten levels for DID and FR1, respectively. The effects of VSD on breaking point values and ethanol consumption during PR was analyzed by a one-way ANOVA, with a between-subjects variable, VSD.

Similarly, all behavioral data were analyzed using a one-way ANOVA with a between-subjects variable (Control and VSD), except for the data of the time spent in the corners, for which a two-way ANOVA was performed with a between subjects variable, VSD, with two levels (Control and VSD) and a within-subjects variable, Sessions, with two levels (Object vs. Social). 

Data are presented as mean ± SEM and a p-value < 0.05 was considered statistically significant. Analyses were performed using SPSS v26. In all cases, post-hoc comparisons were performed with Bonferroni tests.

  1. There was no description of Figure 7 in Result section.

We have corrected the lack of a description.

  1. Please discuss why VSD procedure did not cause body weight changed during stress period in VSD group in the discussion (Figure3)

We did not expect to find changes in the body weight of females exposed to VSD. In our studies using four episodes of social defeat, we have not found changes in the body weight of defeated male mice. We must take into account that our protocol is short (25 minutes) and intermittent (each 72 h). If this protocol is not strong enough to affect the body weight of physically attacked males, it is reasonable to expect no changes in females witnessing the encounters. The lack of corticosterone increments 4 h after the end of the VSD indicates the normalization of the physiological reaction of the females, and points to no further physiological alterations.

We have added the following information to the discussion section:

We did not expect to find any changes in the body weights of females exposed to VSD, as no decrements are usually observed in defeated male mice exposed to four social defeat encounters. We must take into account that our protocol is short (25 minutes) and intermittent (each 72 h). The lack of corticosterone increments 4 h after the end of the VSD indicates the normalization of the physiological reaction of the females and points to the fact that no further physiological alterations should happen.

  1. Please discuss why there was no increase in motivation to obtain ethanol (figure 8d) in the discussion

The PR schedule is complementary to the FR schedules and is indicative of the motivation for seeking the drug, while FR1 assesses the potential liability of a drug and the consumption based on its unconditioned psychopharmacological effects. Our results indicate that VSD females do not show motivational alteration towards reward. Several reasons could explain this lack of effect. The distress induced by the VSD could be less intense than that experienced by physically defeated mice. Additionally, we noticed that non-stressed females made a higher number of active responses during the PR (BP 25) than control male mice (BP less than 15). In fact, defeated male mice showed a BP between 25 and 30.

The clarify this point in the revised manuscript, we have added the following information in the Discussion section:

     In contrast with defeated male mice, VSD females showed similar BP than control females. Several reasons could explain this lack of an effect. We can consider that the distress induced by the VSD could be less intense than that experienced by physically defeated mice. Alternatively, the high number of active responses made by control females during the progressive ratio (BP 25) in comparison with control male mice (BP less than 15) could mask an increase in motivation for ethanol.

Round 2

Reviewer 2 Report

Thank you very for the modifications. 

I still recommend to  analyze the inflammatory response after the stress using ELISA or western blot or flow cytometry, ... and add some results in this study even if it' s a negative results I would bring mechanistic to the study. 

Thank you 

Author Response

I still recommend to  analyze the inflammatory response after the stress using ELISA or western blot or flow cytometry, ... and add some results in this study even if it' s a negative results I would bring mechanistic to the study. 

Although, we would like to have more results (such as other inflammatory markers in more brain areas), we have added the results that we have to date: IL-6 levels in striatum after the last VSD and after the cocaine-induced CPP procedure. The results indicate no changes in VSD female.

We have made the appropriate changes in the Introduction, Material and Methods, Results and Discussion. A new  figure (Figure 9) and 5 references have been added to the manuscript.

Reviewer 4 Report

The authors address all the reviewer's comments carefully and have improved their manuscript considerably. I have no further comments. 

Author Response

The authors address all the reviewer's comments carefully and have improved their manuscript considerably. I have no further comments. 

Thank you for your kind comments

Round 3

Reviewer 2 Report

Thank you for your changes